# Diversity of Endophytic Fungi in Annual Shoots of *Prunus mandshurica* (Rosaceae) in the South of Amur Region, Russia

Eduard V. Nekrasov [1,*], Lyudmila P. Shumilova [1,2], Maria M. Gomzhina [3], Alina V. Aleksandrova [4], Lyudmila Y. Kokaeva [4] and Lyudmila M. Pavlova [2]

[1] Amur Branch, Botanical Garden-Institute FEB RAS, Blagoveshchensk 675000, Russia
[2] Institute of Geology and Nature Management FEB RAS, Blagoveshchensk 675000, Russia
[3] All-Russian Institute of Plant Protection, Pushkin, Saint Petersburg 196608, Russia
[4] Faculty of Biology, Lomonosov Moscow State University, Moscow 119991, Russia
[*] Correspondence: ed_nekrasov@mail.ru

**Abstract:** *Prunus mandshurica* is a rare species of the Russian Far East; it is cultivated for fruits and as an ornamental tree. The objective was to determine the fungi associated with young shoots of the Manchurian apricot, which is an important biotic factor for plant viability and productivity. Fungi were isolated by incubation of shoot fragments (unsterilized or surface-sterilized) on a growth medium and identified according to their cultural and morphological characteristics. *Diaporthe eres* and *Nothophoma quercina* isolates were identified by multilocus phylogenetic analysis (*apn2*, *cal*, *tef1-α*, *tub2* for *D. eres*, and ITS, *rpb2*, *tub2* for *N. quercina*). In total, 12 species (*Alternaria alternata*, *A. tenuissima*, *Aureobasidium pullulans*, *Cladosporium cladosporioides*, *C. herbarum*, *D. eres*, *Epicoccum nigrum*, *Fusarium graminearum*, *F. oxysporum*, *N. quercina*, *Sarocladium strictum*, and *Tripospermum myrti*) and one genus (*Gliocladium* sp.) were found. *Alternaria alternata*, *N. quercina*, and *D. eres* were the most frequent species of the shoots. *Alternaria tenuissima* and *F. oxysporum* were also frequent in some collections, while all other species were rare or occasional in occurrence. Molecular analysis of *D. eres* and *N. quercina* revealed redundancy of some species within the *D. eres* species complex and the genus *Nothophoma*. This is the first report on the fungal inhabitants of asymptomatic shoots of *P. mandshurica*. *Nothophoma quercina* was identified in Russia for the first time.

**Keywords:** biodiversity; endophytic fungi; epiphytic fungi; distribution in shoots; Manchurian apricot; molecular phylogeny; pycnidial fungi; seasonal changes

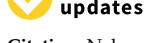



## 1. Introduction

The lifestyles of the fungi associated with plants can be saprophytic, symbiotic, pathogenic, or a combination of these resulting in a number of interactions with their host plants ranging from mutualism to parasitism [1–3]. Fungi colonize both the surface of plant aerial parts and its internal tissues and, thus, can be subdivided into epiphytes and endophytes [4]. While such dividing is simply based on the localization of the fungi, endophytes and epiphytes are usually distinguished from both mycorrhizal and pathogenic fungi [1–4]. The importance of pathogenic and mycorrhizal fungi for agriculture and forestry traditionally attracts the major attention of researchers. However, since the end of the 20th century, the interest in fungal endophytes had also increased [1,5]. By definition, endophytic fungi live inside plant tissues, and do not cause any symptoms of disease in the host [1,6,7]. There is growing experimental evidence indicating the potential involvement of endophytic fungi in plant protection against biotic and abiotic stresses, regulation of plant growth, and productivity [5–8]. Under some conditions, endophytic fungi may become pathogenic for their host plant or adopt a saprotrophic lifestyle upon host decay [9].

Manchurian apricot (*Prunus mandshurica* (Maxim.) Koehne) is distributed in Northeast China, the south of Far Eastern Russia, and Korea. Since the beginning of the apricot

cultivation in Blagoveshchensk (the south of the Amur region, Russia) early in the 20th century, the species has been grown in orchards and private gardens for fruits and planted as an ornamental plant in urban areas. Under favorable conditions, natural renewal occurs and newly grown trees are observed away from the sites of initial planting. Due to its high frost-resistance, the Manchurian apricot is a valuable object for breeding; thus, some cultivars originated from the species have been developed [10]. The species is listed in the *Red Book of the Russian Federation* [11] in the category "Rare". This ensures *P. mandshurica* is a species which is to be protected. Besides climatic factors and anthropogenic pressure, the preservation of the species strongly depends on biotic factors which necessarily include fungi associated with the plant. The mycobiota of apricot as well as other species of *Prunus* is studied mostly regarding fungal pathogenicity on shoots with manifestations of diseases [12–20]. Healthy or asymptomatic shoots and branches have been also studied for endophytic fungi in some *Prunus* species: *Prunus africana* (Hook. F.) Kalkman [21], *P. avium* L. [22,23] and its grafts on different rootstocks [24], *P. cerasus* L. [22], *P. dulcis* [25], and *P. persica* (L.) Batsch. [26,27]. Fungi isolated from visually healthy shoots of fruit plants including apricot (*Prunus armeniaca* L.), cherry (*P. cerasus*), plum (*P. domestica* L.), peach (*P. persica*), apple, and pear were reported in the work [28]. The diversity of fungi in the healthy tissues of the Manchurian apricot remains unstudied. However, micropropagation of the Manchurian apricot revealed a high level of fungal contamination of woody shoots used as explants for in vitro culture initiation [29]. While the used shoots were annual and without obvious symptoms of diseases, the contaminating fungi might arise both from the surface or internal parts of the plant material, thus representing epiphytic or endophytic mycobiota of the apricot.

Apricot annual shoots bear vegetative buds, which give rise to the majority of new shoots and leaves in the next growing season, and all flower buds providing fruiting and seed reproduction of the plant. Therefore, an investigation of the mycobiota of the growing and one-year-old shoots is important for the understanding of the process of colonization and the subsequent impact which fungi may have on vegetative/photosynthetic and generative organs of a plant in the current and following growing seasons.

The aim of this study is to evaluate the fungal diversity in annual shoots of the Manchurian apricot growing in the south of the Amur region. The study is restricted to mycelial fungi and does not include yeasts.

## 2. Materials and Methods

### 2.1. Plant Material, Sampling, and Fungal Isolation

Shoots of *P. mandshurica* were sampled from an abandoned orchard located in the suburb of Blagoveshchensk, Russia (Plodopitomnik, Blagoveshchensk, Amur region, 50.318333° N 127.478333° E) during 2013–2015. At the moment of the study, the age of the initial plantings was at least 40 years without any agricultural care for about 10 years. Field survey of the apricot trees was conducted in August 2014 and August 2016. The trees were inspected through visual observations of their physiological and structural conditions according to the national sanitary regulations [30]. Voucher specimens of *P. mandshurica* (under the synonym name *Armeniaca mandshurica* (Maxim.) B. Skvortz.) were deposited in the herbarium of the Amur Branch of Botanical Garden-Institute FEB RAS (ABGI) (Blagoveshchensk, Russia) under accession numbers ABGI45249–ABGI45250.

The used apricot trees were in the generative stage of development, at least 3 m tall, and included old ones initially planted in the orchard and younger self-seeded plants. Branches without visible symptoms of disease were cut from the lower parts of tree crowns in the morning hours. The branches were processed for microbiological studies and in vitro experiments on the day of collection. Leaves, if present, were removed and discarded before washing. For isolation of fungi, annual twigs or young shoots were detached from the branches and washed in a washing powder solution (0.5% $w/v$) on a magnetic stirrer for 30 min followed by rinsing with tap water at least five times. Surface sterilization was done in two steps: 70% ($v/v$) ethanol for 2 min followed by 0.2% ($w/v$) mercury (II) chloride with

the addition of Triton X-100 (one drop per 50 mL solution) for 10 min. The material was rinsed with sterile distilled water four times. Young green shoots, collected on 10 June 2014, were washed as above but surface-sterilized by a solution of 25% commercial bleach (ca. 5.4% sodium hypochlorite) for 10 min followed by rinsing with sterile water as above [29].

The sterilized twigs and young shoots were used entirely or divided in upper, middle, and basal sections under sterile conditions. Usually, each section was further divided in a node, an adjoined internode, and buds. The basal sections with minute buds and shortened internodes were cut in half longitudinally. In some trials, bud scales were removed. Obtained parts of each twig were placed in a Petri dish with Czapek's agar [31] and left at room temperature. Washed, unsterilized twigs and young shoots as well as untreated ones were also tested for fungal contamination.

In one experiment, dormant twigs (sampling date: 24 February 2015) after the surface sterilization as above or without a sterilization were cut with scissors into pieces about 5 mm long, transferred into a flask, weighed, and added to sterile water (10:1, $v/w$). The content was shaken on a shaker for 15–30 min at room temperature. An aliquot (1 mL) was withdrawn and mixed with sterile water (9 mL) for a 100-fold dilution. A 1000-fold dilution was prepared in a similar way. At each dilution, 10 aliquots (0.1 mL each) were taken and dispersed in 10 Petri dishes with Czapek's agar. The Petri dishes with inoculated media were incubated at room temperature.

Some fungal isolates were obtained in the experiments on micropropagation of the Manchurian apricot after the introduction of the plant into in vitro culture [29]. Dates of plant sampling, parts of the apricot shoots used for the introduction (types of explants), methods of surface sterilization, and nutrient media are shown in Section 3.4, Table 6. The fungal colonies appeared during the first month after the in vitro culture initiation (up to the first subculture) were only taken into account.

The frequency of occurrence (FO) of a fungal species was determined as the percentage of samples (shoots) in which the species was found compared to the total number of samples used in an experiment. Fungal species were considered as occasional (FO < 10%), rare (10 ≤ FO < 30%), frequent (30 ≤ FO < 60%), and dominant (FO ≥ 60%), respectively [32].

*2.2. Morphological Characterization of Fungi*

Each fungal morphotype was isolated into pure cultures by the three-point inoculation method on Czapek's solid medium and allowed to grow for 7–10 days, or 15–30 days for slowly growing species. Other culture media used in the study included potato-sucrose agar (PSA) [31], oatmeal agar (OA) [31], and starvation agar [33]. Isolated strains were identified according to their cultural (colony size and pattern, color in colony surface and reverse, colony surface texture, pigment exudation in medium, exudates, microscopic appearance of formed organs) and morphological characteristics (type, size, and shape of conidia, their formation pattern, a structure of conidiophores, and others) [34–39]. Microscopic slides were prepared in 40% lactic acidand examined with a light microscope (Micromed 1 var. 3-20, OOO "Nablyudatelnye pribory" Saint Petersburg, Russia, or AxioLab.A1, Carl Zeiss MicroImaging GmbH, Gottingen, Germany). Images were taken with an AxioCam ERc5s camera and an AxioVs40 V 4.8.2.0 software (Carl Zeiss MicroImaging GmbH, Gottingen, Germany).

For induction of sporulation in *Epicoccum nigrum* Link, the strain was grown on starvation agar at 25 °C for 10 days. The fungal strains which failed to sporulate following incubation (for 30 days) were considered as mycelia sterilia. For morphological studies of pycnidial fungi, the strains were grown in PSA and OA. The inoculated Petri dishes were incubated in the dark at 20–22 °C for 7 days. On the second week of growth, the dishes were irradiated with UV light (low-pressure fluorescent erythemal discharge lamp LE-30 with maximum irradiation at 310–320 nm, OOO "NIIIS imeni A.N. Lodygina", Saransk, Russia) for 13 h per day [40]. Morphological characteristics were recorded after 14 days of colony growth. Observations and measurements of 100 conidia for each isolate were conducted with an Olympus SZX16 stereomicroscope (Olympus, Tokyo, Japan) and an

Olympus BX53 microscope. Images were captured with a PROKYON camera (Jenoptik, Jena, Germany).

Strains were preserved in agar slant tubes (Czapek's agar or PSA) at +4 °C and are stored in the collections of the Laboratory of Biogeochemistry (Institute of Geology and Nature Management, Blagoveshchensk, Russia) and Laboratory of Mycology and Phytopathology (All-Russian Institute of Plant Protection (VIZR), Saint Petersburg, Russia). Names of fungal species and families are given according to the databases [41,42].

### 2.3. Molecular Characterization and Phylogenetic Analysis of Pycnidial Fungi

In total, fourteen isolates of the pycnidial fungi (nine of Didymellaceae sp. and five of *Diaporthe* sp.), which had been difficult to identify by morphology, were investigated using molecular techniques as described below.

Fungal mycelia were obtained from the cultures grown on PSA and the biomass was macerated with 0.3 mm glass sand on an MM400 mixer mill (Retsch GmbH, Haan, Germany). Genomic DNA was extracted according to the standard CTAB/chloroform protocol [43].

The isolates were preliminary identified to a genus level by internal transcribed spacer (ITS) regions sequencing. For species identification, DNA sequences encoding partial β-tubulin (*tub2*) and partial RNA polymerase II second largest subunit (*rpb2*) were amplified and sequenced for the Didymellaceae isolates; and those encoding partial DNA-lyase (*apn2*), *tub2*, calmodulin (*cal*), and translation elongation factor 1-α (*tef1-α*) for the *Diaporthe* isolates.

The primers ITS1F [44] and ITS4 [45], apn2fw2 and apn2rw2 [46], CAL-228F and CAL-737R [47], βtub2Fw and βtub4Rd [48], EF1-728F and EF1-986R [47], fRPB2-5F2 [49], and fRPB2-7cR [50] were used to amplify the ITS region, partial *apn2*, *tub2*, *cal*, *tef1-α*, and *rpb2*, respectively. Polymerase chain reaction (PCR) was done in the volume of 25μL in the presence of deoxynucleotide triphosphates (dNTPs, 200 μM), forward and reverse primers (0.5 μM each), Taq DNA polymerase (5 U/μL), 10× PCR buffer with $MgCl_2$ and $NH_4Cl$, and 1–10 ng of total genomic DNA. PCR was carried out in a thermocycler Bio-Rad C1000 Touch (Bio-Rad Laboratories, Hercules, CA, USA) under conditions as follows: 95 °C for 5 min followed by 35 cycles at 92 °C for 50 s, then at 55 °C for 40 s (ITS1F/ITS4), or 54 °C for 40 s (apn2fw2 and apn2rw2), or 56 °C for 30 s (CAL-228F/CAL-737R), or 52 °C for 40 s (βtub2Fw/βtub4Rd), or 55 °C for 60 s (EF1-728F/EF1-986R), followed by 72 °C for 75 s, and a final extension at 72 °C for 5 min. The gene *rpb2* was amplified by touchdown PCR: all steps were the same as described above, but the annealing temperature consequently decreased from 5 cycles at 60 °C for 40 s and 5 cycles at 58 °C for 40 s to 30 cycles at 54 °C for 40 s. The resulting PCR products were run in a 1% agarose gel and stained with ethidium bromide. Amplicons were purified according to the method [51] and sequenced by Sanger's method [52] on an ABI Prism 3500 Genetic Analyzer (Applied Biosystems, Thermo Fisher Scientific, Waltham, MA, USA) using a BigDye Terminator v3.1 Cycle Sequencing kit (Applied Biosystems, Thermo Fisher Scientific, Waltham, MA, USA) according to the manufacturer's protocols. The obtained nucleotide sequences of the *apn2*, *tub2*, *cal*, *tef1-α*, and *rpb2* genes were deposited in the GenBank database with corresponding accession numbers (Table 1).

Sequences were assembled using Vector NTI advance v. 11.0 (Invitrogen, Carlsbad, CA, USA) and aligned with ClustalX 1.8 [53]. The alignments were optimized with Molecular Evolutionary Genetics Analysis 10 (MEGA X, [54]) and concatenated using SequenceMatrix [55]. Sequences of representative *Diaporthe* and Didymellaceae strains and type species were retrieved from GenBank (Table 1).

Two different datasets were made to implement two phylogenetic analyses. The first set for the *Diaporthe* isolates was based on the combined *apn2*, *tub2*, *cal*, and *tef1-α* sequences for five studied isolates, with *Diaporthe citri* F.A. Wolf (AR3405 = CBS 135422) used as an outgroup. The second set included the combined data from the ITS, *tub2*, and *rpb2*

sequences for the nine studied Didymellaceae isolates with *Macroventuria anomochaeta* Aa (CBS 525.71) used as an outgroup.

Phylogenetic analysis of combined aligned data consisted of maximum likelihood (ML), maximum parsimony (MP), and Bayesian inference (BI). Both ML and MP analyses were performed with MEGA X. Bayesian inference was carried out by Mr. Bayes v. 3.2.1. in ARMADILLO v. 1.1 [56]. The ML analyses were performed on a neighbor-joining starting tree automatically generated by the software. Nearest neighbor interchange (NNI) was used as a heuristic method for tree inference and 1000 bootstrap replicates were performed. MEGA X was also used to determine the best nucleotide substitution model to be used for building the ML trees. Bootstrap values with 1000 replications were calculated for tree branches.

The MP analyses were performed using the heuristic search option with 100 random taxon additions and the subtree pruning regrafting (SPR) method as a branch-swapping algorithm. All characters were unordered and of equal weight, and gaps were treated as missing data. Maxtrees were set to 100 and branches of zero length were collapsed. Clade stability was assessed using a bootstrap analysis with 1000 replicates. The BI analyses were performed employing a Markov chain Monte Carlo sampling (MCMC) method. The general time-reversible model of evolution including estimation of invariable sites and assuming a gamma distribution with six rate categories was used for the BI analyses. Four MCMC chains were run simultaneously starting from random trees for 1000 generations and sampled every 10th generation for a total of 10,000 trees.

The model test in MEGA X determined that the TN+F+G4 model was most appropriate for *apn2*, *tub2*, *cal*, and *tef1-α* according to Bayesian information criteria (BIC). The TN+G4 model was the most suitable nucleotide substitution model for ITS, *tub2*, and *rpb2* according to BIC. Bootstrap values with 1000 replications were calculated for tree branches.

**Table 1.** List of studied and reference species and strains.

| Species | StrainNumber | GenBank Accession Number | | | | | |
|---|---|---|---|---|---|---|---|
| | | *tub2* | *tef1-α* | *cal* | *apn2* | *rpb2* | ITS |
| *Diaportheambigua* | CBS 114015 | KC343978 | KC343736 | KC343252 | - | - | - |
| *D.amygdali* | CBS 126679 | KC343990 | KC343748 | KC343264 | - | - | - |
| *D. citri* | AR3405 = CBS 135422 | KC344020 | KC343778 | KC843157 | KJ380981 | - | - |
| *D. eres* | **MF-Pm-1a** | **MZ671975** | **MZ671970** | **MZ671923** | **MZ671918** | - | **MZ646151** |
| *D. eres* | **MF-Pm-2a** | **MZ671976** | **MZ671971** | **MZ671924** | **MZ671919** | - | **MZ646152** |
| *D. eres* | **MF-Pm-3a** | **MZ671977** | **MZ671972** | **MZ671925** | **MZ671920** | - | **MZ646153** |
| *D. eres* | **MF-Pm-4a** | **MZ671978** | **MZ671973** | **MZ671926** | **MZ671921** | - | **MZ646154** |
| *D. eres* | **MF-Pm-5a** | **MZ671979** | **MZ671974** | **MZ671927** | **MZ671922** | - | **MZ646155** |
| *D. eres* | MF-Ha18-001 | MK033490 | MK039422 | MZ671934 | MZ671915 | - | - |
| *D. eres* | MF-Ha18-002 | MK033491 | MK039423 | MZ671935 | MZ671916 | - | - |
| *D. eres* | MF-Ha18-003 | MW008495 | MW008506 | MZ671931 | MZ671917 | - | - |
| *D. eres* | CBS 495.72 | KC343975 | KC343733 | KC343249 | KJ380963 | - | - |
| *D. eres* | CBS 146.46 | KC343976 | KC343734 | KC343250 | KJ380969 | - | - |
| *D. eres* | CFCC 50469 | KT733020 | KT733016 | KT732997 | - | - | - |
| *D. eres* | CFCC 52562 | MH121579 | MH121539 | MH121421 | - | - | - |
| *D. eres* | CBS 121004 | KC344102 | KC343860 | KC343376 | KJ380976 | - | - |
| *D. eres* | CGMCC 3.17081 | KF576306 | KF576257 | - | - | - | - |
| *D. eres* | CBS 146962 | MN136190 | MN136153 | MN136129 | MN136122 | - | - |
| *D. eres* | CBS 587.79 | KC344121 | KC343879 | KC343395 | KJ380975 | - | - |
| *D. eres* | CFCC 51632 | KY228893 | KY228887 | KY228877 | - | - | - |
| *D. eres* | DNP128 | JX275438 | JX275401 | JX197430 | - | - | - |
| *D. eres* | CBS 139.27 | KC344015 | KC343773 | KC343289 | KJ380974 | - | - |
| *D. eres* | CPC 28262 | MG281190 | MG281538 | MG281712 | - | - | - |
| *D. eres* | CFCC 52567 | MH121584 | MH121544 | MH121426 | - | - | - |
| *D. eres* | DP0667 | KC843229 | KJ210548 | KC843155 | KJ380923 | - | - |
| *D. eres* | CGMCC 3.17084 | KF576291 | KF576245 | - | - | - | - |
| *D. eres* | AR3672 = MAFF625034 = CBS 116964 | KJ420819 | JQ807418 | KJ435023 | KJ380937 | - | - |
| *D. eres* | AR5211 = CBS 138596 | KJ420828 | KJ210559 | KJ435043 | KJ380977 | - | - |
| *D. eres* | CGMCC 3.17089 | KF576291 | KF576242 | - | - | - | - |
| *D. eres* | CGMCC 3.15181 | KF576312 | KC153087 | KT459461 | - | - | - |
| *D. eres* | DAOMC 250563 | KU574616 | KU552022 | - | KU552020 | - | - |
| *D. eres* | MFLUCC 16-0113 | KU557587 | KU557631 | KU557611 | - | - | - |
| *D. eres* | CBS 144. 27 | KC344112 | KC343870 | KC343386 | KJ380973 | - | - |

**Table 1.** *Cont.*

| Species | StrainNumber | GenBank Accession Number | | | | | |
|---|---|---|---|---|---|---|---|
| | | *tub2* | *tef1-α* | *cal* | *apn2* | *rpb2* | ITS |
| *D. eres* | CFCC 52590 | MH121604 | MH121567 | MH121443 | - | - | - |
| *D. eres* | CBS 138897 | KP004507 | - | - | - | - | - |
| *D. eres* | CBS 338.89 | KC344120 | KC343878 | KC343394 | KJ380978 | - | - |
| *D. eres* | MFLU 17-0646 | MG843877 | MG829270 | MG829274 | - | - | - |
| *D. eres* | CBS 160.32 | KC344196 | KC343954 | KC343470 | KJ380968 | - | - |
| *D. eres* | CAA1001 | MT309458 | MT309432 | MT309449 | - | - | - |
| *D. eres* | AR5193 * | KJ420799 | KJ210550 | KJ434999 | KJ380958 | - | - |
| *D. eres* | AR5196 | KJ420817 | KJ210554 | KJ435006 | KJ380932 | - | - |
| *D. eres* | DP0438 | KJ420816 | KJ210553 | KJ435016 | KJ380935 | - | - |
| *D. eres* | FAU483 | KJ420827 | JQ807422 | KJ435022 | KJ380933 | - | - |
| *D. eres* | DAN001A | KJ420781 | KJ210540 | KJ434994 | KJ380914 | - | - |
| *D. eres* | DAN001B | KJ420782 | KJ210541 | KJ434995 | KJ380915 | - | - |
| *D. eres* | AR3519 | KJ420789 | KJ210547 | KJ435008 | KJ380922 | - | - |
| *D. eres* | FAU570 | KJ420794 | JQ807410 | KJ435025 | KJ380926 | - | - |
| *D. eres* | AR3723 | KJ420793 | JQ807351 | KJ435024 | KJ380941 | - | - |
| *D. eres* | AR3560 | KJ420795 | KJ210551 | KJ435011 | KJ380939 | - | - |
| *D. eres* | AR5224 | KJ420802 | KJ210555 | KJ435036 | KJ380961 | - | - |
| *D. eres* | AR5231 | KJ420818 | KJ210549 | KJ435038 | KJ380936 | - | - |
| *D. eres* | AR5223 | KJ420830 | KJ210549 | KJ435000 | KJ380938 | - | - |
| *D. eres* | DLR12a | KJ420783 | KJ210542 | KJ434996 | KJ380916 | - | - |
| *D. eres* | AR4369 | KJ420813 | JQ807366 | KJ435005 | KJ380953 | - | - |
| *D. eres* | MF-Vm17-001 | MZ054675 | MZ054665 | - | MZ054647 | - | - |
| *D. eres* | MF-Vm17-008 | MZ054676 | MZ054666 | - | MZ054648 | - | - |
| *D. eres* | MF-Vm17-009 | MZ054677 | MZ054667 | - | MZ054649 | - | - |
| *D. eres* | MF-Vm17-019 | MZ054678 | MZ054669 | - | MZ054650 | - | - |
| *D. eres* | MF-Vm17-030 | MZ054679 | MZ054670 | - | MZ054651 | - | - |
| *D. eres* | CFCC 52576 | MH121593 | MH121553 | MH121432 | - | - | - |
| *D. eres* | AR3538 = CBS 109767 | KC344043 | KC343801 | KC343317 | KJ380940 | - | - |
| *D. eres* | FAU506 | KJ420792 | JQ807403 | KJ435012 | KJ380925 | - | - |
| *D. eres* | FAU532 | KJ420815 | JQ807408 | KJ435015 | KJ380934 | - | - |
| *D. foeniculina* | CBS 111553 * | KC344069 | KC343827 | KC343343 | - | - | - |
| *D. malorum* | CBS 142383 = CAA734 * | KY435668 | KY435627 | KY435658 | - | - | - |
| *D. sennicola* | CFCC 51634 * | KY228889 | KY228883 | KY228873 | - | - | - |
| *Macroventuria anomochaeta* | CBS 525.71 * | GU237545 | - | - | - | GU456346 | MH860249 |

**Table 1.** *Cont.*

| Species | StrainNumber | GenBank Accession Number | | | | | |
|---|---|---|---|---|---|---|---|
| | | *tub2* | *tef1-α* | *cal* | *apn2* | *rpb2* | ITS |
| *Nothophoma acaciae* | CBS 143404 * | MG386167 | - | - | - | MG386144 | MG386056 |
| *N. anigozanthi* | CBS 381.91 * | GU237580 | - | - | - | KT389655 | GU237852 |
| *N. arachidis-hypogaeae* | CBS 125.93 * | GU237583 | - | - | - | KT389656 | GU237771 |
| *N. brennandiae* | CBS 145912 * | MN824753 | - | - | - | MN824604 | MN823579 |
| *N. garlbiwalawarda* | BRIP 69595 * | - | - | - | - | MN604937 | MN5676786 |
| *N. eucalyptigena* | CBS 142535 * | KY979935 | - | - | - | KY979852 | KY979771 |
| *N. gossypiicola* | CBS 377.67 | GU237611 | - | - | - | KT389658 | GU237845 |
| *N. infossa* | CBS 123395 * | FJ427135 | - | - | - | KT389659 | FJ427025 |
| *N. infuscata* | CBS 121931 * | MT005662 | - | - | - | MT018203 | MN973559 |
| *N. macrospora* | CBS 140674 * | LN880539 | - | - | - | LT593073 | LN880536 |
| *N. naiawu* | BRIP 69583 * | - | - | - | - | MN604938 | MN5676787 |
| *N. nullicana* | CPC 32330 * | MG386165 | - | - | - | MG386143 | NR_156665 |
| *N. pruni* | JZB380017 | MH853670 | - | - | - | MH853663 | MH827006 |
| *N. pruni* | JZB380015 | MH853668 | - | - | - | MH853661 | MH827004 |
| *N. pruni* | MFLUCC 18-1601 | MH853669 | - | - | - | MH853662 | MH827005 |
| *N. pruni* | JZB380038 | MN991303 | - | - | - | MN991306 | MN533798 |
| *N. pruni* | MFLUCC: 18-1600 * | MH853671 | - | - | - | MH853664 | MH827007 |
| *N. quercina* | CBS 633.92 * | GU237609 | - | - | - | KT389657 | GU237900 |
| *N. quercina* | **MF-Pm-6a** | **MZ671980** | - | - | - | **MZ671944** | **MZ646156** |
| *N. quercina* | **MF-Pm-7a** | **MZ671981** | - | - | - | **MZ671945** | **MZ646157** |
| *N. quercina* | **MF-Pm-8a** | **MZ671982** | - | - | - | **MZ671946** | **MZ646158** |
| *N. quercina* | **MF-Pm-9a** | **MZ671983** | - | - | - | **MZ671947** | **MZ646159** |
| *N. quercina* | **MF-Pm-10a** | **MZ671984** | - | - | - | **MZ671948** | **MZ646160** |
| *N. quercina* | **MF-Pm-12a** | **MZ671986** | - | - | - | **MZ671950** | **MZ646162** |
| *N. quercina* | **MF-Pm-13a** | **MZ671987** | - | - | - | **MZ671951** | **MZ646163** |
| *N. quercina* | **MF-Pm-14a** | **MZ671988** | - | - | - | **MZ671952** | **MZ646164** |
| *N. quercina* | **MF-Pm-15a** | **MZ671989** | | | | **MZ671953** | **MZ646165** |
| *N. quercina* | JZB380007 | MZ646165 | - | MZ671989 | - | - | MZ671953 |
| *N. quercina* | JZB380009 | KY887673.1 | - | KY887679 | - | - | KY887677 |
| *N. quercina* | MFLUCC 18–1588 | MH827008 | - | MH853672 | - | - | MH853665 |
| *N. quercina* | CGMCC:3.19246 | MK088574 | - | MK088595 | - | - | MK088588 |

**Table 1.** *Cont.*

| Species | StrainNumber | GenBank Accession Number | | | | | |
|---------|--------------|------|--------|-----|------|------|------|
| | | *tub2* | *tef1-α* | *cal* | *apn2* | *rpb2* | ITS |
| *N. quercina* | JZB380039 | MN533799 | - | MN537423 | - | - | MN537426 |
| *N. quercina* | LC12187 | MK088575 | - | MK088596 | - | - | MK088589 |
| *N. quercina* | XJAKS05 | KX225387 | | | | | KX645664 |
| *N. quercina* | ZQ202004002 | MW883394 | - | - | - | MW883395 | MW541930 |
| *N. quercina* | EAH 2 | MW330391 | - | - | - | MW330390 | MW325676 |
| *N. quercina* | Ph1 | MK522081 | - | - | - | - | MK522080 |
| *N. quercina* | JZB380108 | ON351014 | - | - | - | ON350993 | ON316870 |
| *N. quercina* | JZB380106 | ON351012 | - | - | - | ON350991 | ON316868 |
| *N. quercina* | 469E | - | - | - | - | - | MZ078709 |
| *N. quercina* | Hz4-1 | ON961030 | - | - | - | ON996909 | ON429028 |
| *N. quercina* | CBS 159.37 | MN984016 | - | - | - | - | MN973004 |
| *N. quercina* | MF-32.61 | - | - | - | - | - | KY552963 |
| *N. spiraeae* | CFCC 53928 * | MN879295 | - | - | - | MN879292 | MN737833 |
| *N. variabilis* | CBS 142457 * | LT593008 | - | - | - | LT593078 | LT592939 |

Notes. The studied isolates and new generated sequences are in bold. Ex-type isolates are indicated with an asterisk. Acronyms of culture collection: AR, DAN, DLR, DP, FAU—isolates in the culture collection of Systematic Mycology and Microbiology Laboratory, USDA-ARS, Beltsville, MD, USA; BRIP—Plant Pathology Herbarium, Department of Employment, Economic, Development and Innovation, Dutton Park, QLD, Australia; CAA—Personal Culture Collection Artur Alves, University of Aveiro, Aveiro, Portugal; CBS—Westerdijk Fungal Biodiversity Institute, Utrecht, The Netherlands; CFCC—China Forestry Culture Collection Center, Beijing, China; CGMCC—China General Microbiological Culture Collection Center, Beijing, China; CPC—Culture Collection of Pedro Crous; DAOM—Canadian Collection of Fungal Cultures, Ottawa, ON, Canada; JZB—Beijing Academy of Agriculture and Forestry Sciences Culture Collection, Beijing, China; LC—Personal Culture Collection Lei Cai, State Key Laboratory of Mycology, Institute of Microbiology, Chinese Academy of Sciences, Beijing, China; MAFF—MAFF Genebank Project, Ministry of Agriculture, Forestry and Fisheries, Tsukuba, Japan; MF—Collection of Pure Cultures of the Laboratory of Mycology and Phytopathology, All-Russian Institute of Plant Protection, VIZR, Saint Petersburg, Russia; MFLU—Herbarium of Mae Fah Luang University, Chiang Rai, Thailand; MFLUCC—Mae Fah Luang University Culture Collection, Chiang Rai, Thailand; SAUCC—Shandong Agricultural University Culture Collection, Taian, Shandong, China; ZJUP—Zhejiang University, Hangzhou, Zhejiang, China.

## 3. Results

### 3.1. Sanitary Conditions of the Apricot Trees

The conditions of the trees were strongly dependent on their age. Most of the trees were in a satisfactory condition. A significant portion of the old trees (about 30% of 55 inspected trees), which were growing since the establishment of the orchard, were in a poor condition or dead. The major defects were wood rots, freeze-cracked trunks and limbs, branch dieback, and gummosis. Self-seeded plants (15 inspected trees), which were younger and grew up outside the initial plantation, were in a better condition. Bark necrosis and trunk rots were mostly occasional, whilst branch dieback was less pronounced.

### 3.2. Identification of Isolates by Morphological Features

In total, 332 fungal isolates were obtained from young and annual shoots of *P. mandshurica*. According to their cultural and morphological characteristics, nine genera were identified which represented nine families: Capnodiaceae, Cladosporiaceae, Diaporthaceae, Didymellaceae, Hypocreaceae, Nectriaceae, Pleosporaceae, Dothioraceae, and Sarocladiaceae. A total of 215 isolates were identified to the species level: *Alternaria alternata* (Fr.) Keissl. (one hundred and ten isolates), *A. tenuissima* (Kunze) Wiltshire (thirty-one isolates), *Aureobasidium pullulans* (de Bary et Löwenthal) G. Arnaud (twelve isolates), *Cladosporium cladosporioides* (Fresen.) G.A. de Vries (ten isolates), *Cladosporium herbarum* (Pers.) Link (one isolate), *Epicoccum nigrum* (fifteen isolates), *Fusarium graminearum* Schwabe (two isolates), *Fusarium oxysporum* Schltdl. (thirty isolates), *Sarocladium strictum* (W. Gams) Summerb. (one isolate), and *Tripospermum myrti* (Lind) S. Hughes (three isolates). One isolate was identified to the genus level (*Gliocladium* sp.). Pycnidia-forming fungi were found to belong to *Diaporthe* sp. (forty-two isolates) and Didymellaceae sp. (seventy-four isolates). Four isolates of sterile mycelia were found. Cultural and morphological characteristics of the isolated fungi are given in Supplementary Materials (Morphological Characterization of Fungi).

### 3.3. Molecular Characterization and Phylogenetic Analysis of Didymellaceae sp. and Diaporthe sp. Isolates

*Diaporthe* sp. and Didymellaceae sp. could not be identified to the species level only by the morphological features (Supplementary Materials, Morphological Characterization of Fungi); therefore, the molecular approach was applied for their identification.

Five *Diaporthe* sp. isolates were preliminary identified as members of the *Diaporthe eres* Nitschke species complex by sequencing of the ITS locus. Multilocus phylogenetic analysis of four partial protein-coding genes (*apn2*, *tub2*, *cal*, and *tef1-α*) included sequences of the five studied isolates and sixty-one reference strains and species from the *D. eres* species complex with the outgroup sequences of *D. citri* (CBS 135422) (Table 1).

After trimming at both ends of the *apn2*, *tub2*, *cal*, and *tef1-α* blocks, the total alignment block had a length of 1849 characters including gaps (732 for *apn2*, 422 for *tub2*, 389 for *cal*, and 306 for *tef1-α*). The number of unique site patterns per genome locus was 36 (4.9%), 79 (18.7%), 76 (19.5%), and 114 (37.3%) for the *apn2*, *tub2*, *cal*, and *tef1-α* blocks, respectively. The topology and branching order of ML, MP, and Bayesian phylogenetic trees as well the composition of the phylogenetic clades were similar. Based on the *apn2*, *tub2*, *cal*, and *tef1-α* phylogeny, the isolates clustered into a distinct phylogenetic clade with high bootstrap support. This clade included 55 representative *D. eres* strains together with the ex-epitype strain AR5193. The clade had high statistical support (MLBS 100%, MPBS 100%). The ML tree inferred from the *apn2*, *tub2*, *cal*, and *tef1-α* sequences is shown in Figure 1.

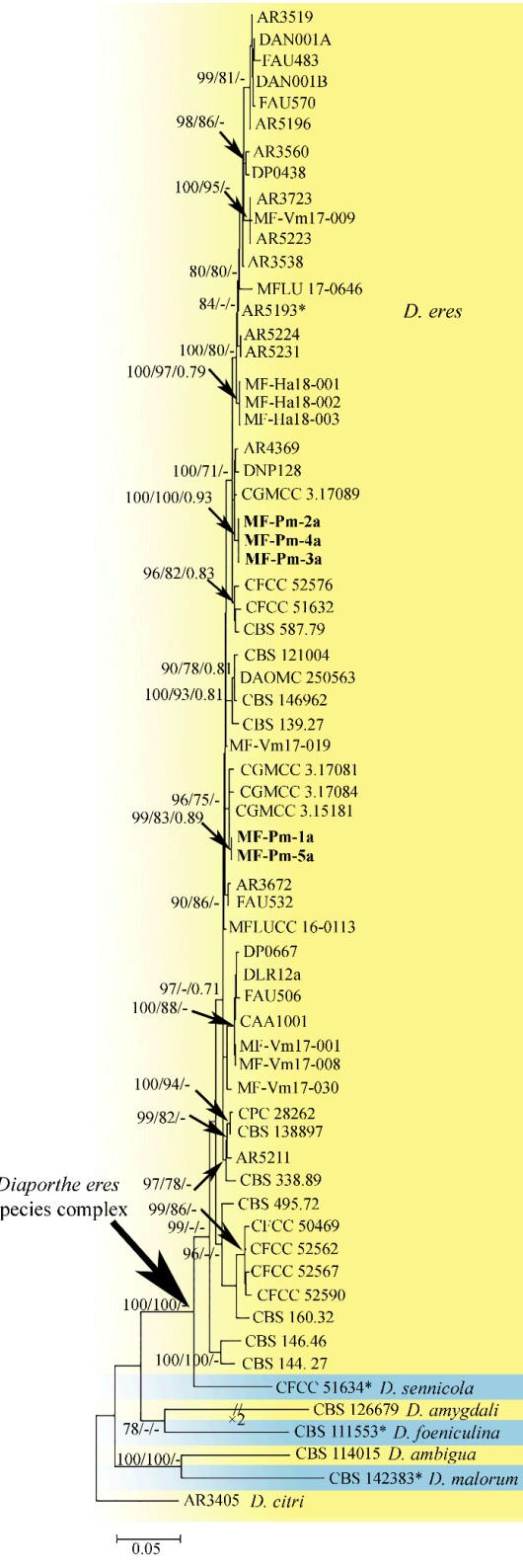

**Figure 1.** Phylogenetic tree of the *Diaporthe eres* species complex inferred from a maximum likelihood (ML) analysis based on a concatenated alignment of *apn2*, *tub2*, *cal*, and *tef1-α*. The ML bootstrap support values (MLBS ≥ 75), MP bootstrap support values (MPBS ≥ 75), and Bayesian posterior probabilities (BPP ≥ 0.75) are given at the nodes (MLBS/MPBS/BPP). Strains isolated in this study are indicated in bold. The ex-type strains are indicated with an asterisk.

The second combined dataset for the ITS, *tub2*, and *rpb2* sequences was analyzed to identify nine Didymellaceae sp. isolates and infer the intraspecific relationships within the *Nothophoma* species with the outgroup sequences of *M. anomochaeta* (Table 1). After trimming at both ends of the ITS, *tub2*, and *rpb2* blocks, the total alignment block had a length of 1319 characters including gaps (434 for ITS, 265 for *tub2*, and 620 for *rpb2*). The number of informative site patterns per genome locus was 23 (5.3%), 33 (12.5%), and 371 (59.8%), respectively. The topology and branching order of ML, MP, and Bayesian phylogenetic trees as well the composition of the phylogenetic clades were similar. Based on the ITS, *tub2*, and *rpb2* phylogeny, the strains clustered into one distinct well-supported (MLBS 100%, MPBS 98%, BBP 0.99) phylogenetic clade with high bootstrap support with the type and representative *Nothophoma brennandiae* Hern.-Restr., L.W. Hou, L. Cai & Crous, *Nothophoma pruni* Chethana, J.Y. Yan, X.H. Li & K.D. Hyde, *Nothophoma quercina* (Syd. & P. Syd.) Qian Chen & L. Cai, and *Nothophoma spiraeae* L.X. Zhang & X.L. Fan strains. The ML tree inferred from the ITS, *tub2*, and *rpb2* sequences is shown in Figure 2. As a result of the molecular phylogeny assessment, we consider four species (*N. brennandiae*, *N. pruni*, *N. quercina*, *N. spiraeae*) and their strains as members of the *N. quercina* species complex. Thus, the nine Didymellaceae strains isolated in our study were identified as *N. quercina*.

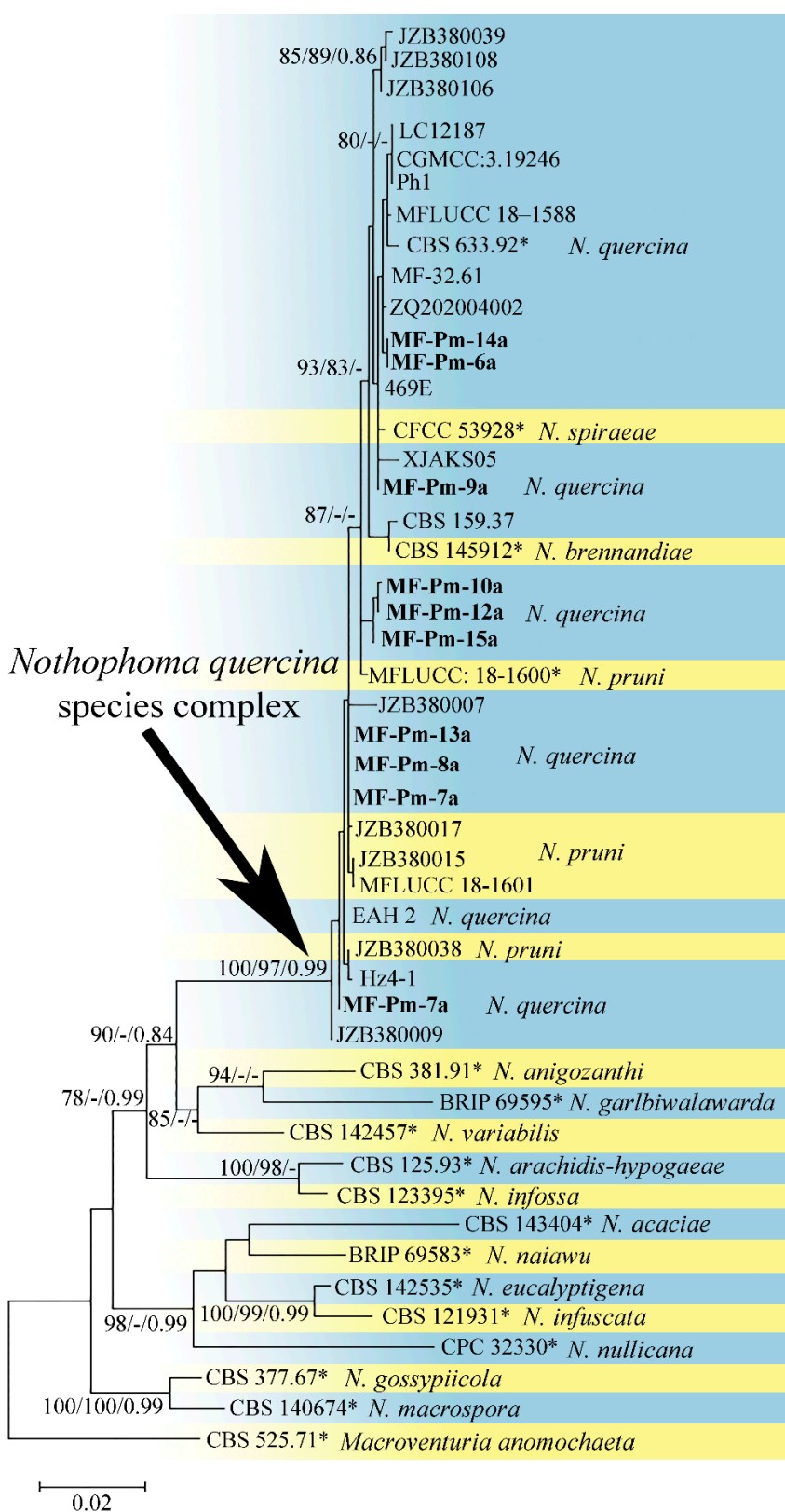

**Figure 2.** Phylogenetic tree of the *Nothophoma* species inferred from a maximum likelihood (ML) analysis based on a concatenated alignment of ITS, *tub2*, and *rpb2*. The ML bootstrap support values (MLBS ≥ 75), MP bootstrap support values (MPBS ≥ 75), and Bayesian posterior probabilities (BPP ≥ 0.75) are given at the nodes (MLBS/MPBS/BPP). Strains isolated in this study are indicated in bold. The ex-type strains are indicated with an asterisk.

### 3.4. Distribution of Fungi in the Apricot Shoots

The distribution of fungi was investigated in the upper, middle, and basal sections of annual shoots during the dormant period in November 2013–March 2014. Almost all the shoots tested in that period were colonized by fungi, although sections and parts differed in the degree of fungal infection (Table 2). Upper (46–90% of all tested shoots) and middle (50–85%) sections were more contaminated than the basal sections (39–70%) of the same shoots. Buds and nodes were the most infected parts of the shoots (23–70% and 15–90%, respectively), while the adjoined internodes were less contaminated (0–40%). When bud scales were separated, fungi were associated with the scales, and the inner parts were mostly free of fungal infection (Table 2).

**Table 2.** Fungal colonization of the dormant annual twigs (November–March) and the newly grown shoots (June) of the Manchurian apricot in 2013–2014.

| | Date of Collection | | | |
| --- | --- | --- | --- | --- |
| | 8 November 2013 | 10 January 2014 | 25 March 2014 | 10 June 2014 |
| Number of tested trees/shoots | 2/13 + 3 [1] | 3/10 + 10 [1] | 5/13 [2] | 5/21 [3] |
| Total infected shoots, % | 100 | 100 | 84.6 | 100 |
| Infected upper sections | | | | |
|    shoots, % | 46.2 | 90 | 61.5 | 85.7 |
|    buds, % | 23.1 | 70 | 61.5 | n.d. |
|    nodes, % | 15.4 | 90 | n.d. | 31.6 |
|    internodes, % | 7.7 | 20 | 15.4 | n.d. |
| Infected middle sections | | | | |
|    shoots, % | 84.6 | 50 | 69.2 | 100 |
|    buds, % | 69.2 | 40 | 61.5 | n.d. |
|    nodes, % | 30.8 | 30 | n.d. | 36 |
|    internodes, % | 0 | 10 | 30.8 | n.d. |
| Infected basal sections | | | | |
|    shoots, % | 61.5 | 70 | 38.5 | 100 |
|    buds, % | 30.8 | 70 | 30.8 | n.d. |
|    nodes, % | 46.2 | 40 | n.d. | 49.4 |
|    internodes, % | 0 | 40 | 30.8 | n.d. |
| Infected inner parts of buds (without scales), % | 0 | 10 | 5.1 | n.d. |
| Infected bud scales, % | n.d. | 100 | 46.2 | n.d. |

Notes. The shoots were collected from predominantly old apricot trees. The dormant shoots were sterilized in the two steps using 70% ethanol and 0.2% mercury chloride. The newly grown shoots were surface disinfected with the solution of 25% commercial bleach. [1] The first number indicates the number of tested trees; the numbers after slash indicate the number of shoots, which were surface-sterilized and cut into sections, and the number (after "+") of the shoots which were used for the separation of buds. [2] Only buds and internodes were investigated for fungi. The scales were removed from the buds. [3] The shoots were divided into upper, middle, and basal sections; nodes including buds were separated from each section. Abbreviation: n.d.—not determined.

The species composition of strains isolated from the surface-sterilized and unsterilized shoots in the dormant stage were not different. The fungi included seven species (Table 3): *A. alternata*, *A. tenuissima*, *C. cladosporioides*, *E. nigrum*, *F. oxysporum*, *N. quercina*, and *D. eres*. The dominant and frequent species were *A. alternata* and *D. eres* in all experiments with the sterilized shoots, but *D. eres* was found only in March in the case of the unsterilized shoots. *Nothophoma quercina* was also a dominant or frequent species, but it was not isolated from the shoots (both sterilized and unsterilized) collected in January. *Fusarium oxysporum* was frequent for the sterilized shoots in November, but it was not isolated from the surface-sterilized shoots collected in January and March, though its sporadic colonies were detected in the case of the unsterilized shoots (Table 3). *Epicoccum nigrum* was isolated in January

both from the treated and untreated shoots. The differences between surface-sterilized and unsterilized shoots can be associated with fungal localization on the shoot surface or in its internal tissues as with sensitivity of the fungi to the sterilization agents.

**Table 3.** Species composition and a frequency of occurrence of fungi isolated from the dormant annual twigs (November–March) and the newly grown shoots (June) of the Manchurian apricot in 2013–2014.

| Fungi | Date of Collection | | | |
|---|---|---|---|---|
| | 8 November 2013 | 10 January 2014 | 25 March 2014 | 10 June 2014 |
| Surface-sterilized shoots | | | | |
| *Alternaria alternata* | F | D | F | O |
| *A. tenuissima* | — | — | F | F |
| *Cladosporium cladosporioides* | R | O | — | O |
| *Diaporthe eres* | D | F | F | R |
| *Epicoccum nigrum* | — | R | — | R |
| *Fusarium oxysporum* | F | — | — | O |
| *Nothophoma quercina* | F | — | D | F |
| Unsterilized shoots | | | | |
| *A. alternata* | + | + | + | + |
| *A. tenuissima* | — | — | + | + |
| *C. cladosporioides* | + | + | — | + |
| *D. eres* | — | — | + | — |
| *E. nigrum* | — | + | — | + |
| *Fusarium graminearum* | — | — | — | + |
| *F. oxysporum* | + | + | + | — |
| *N. quercina* | + | — | + | + |
| Light-colored sterile mycelium | — | — | — | + |
| Dark-colored sterile mycelium | + | — | — | — |

Abbreviations: O—occasional (FO < 10%), R—rare (10 $\leq$ FO < 30%), F—frequent (30 $\leq$ FO < 60%), and D—dominant (FO $\geq$ 60%); "+"—detected; "−"—not detected. See Table 2 for other details.

Fungal colonization of the young green shoots collected from the same trees was investigated by the end of their growth (10 June 2014). The mild procedure used for their surface sterilization (see Materials and Methods) resulted in the 100% infection of the shoots (Table 2). The nods of the basal sections were more infected (49%) as compared to the middle (36%) and upper (32%) sections (Table 2). The strains isolated from the surface-sterilized young shoots represented the same seven species found for the dormant shoots (Table 3). The frequent species were *A. tenuissima* and *N. quercina*, while *D. eres* was rare. Other species were also rare and occasional in the young shoots. *Fusarium graminearum* (Oudem.) Wollenw. and a light-colored sterile mycelium were isolated from unsterilized young shoots in addition to the seven species found for the surface-sterilized shoots (Table 3). Besides the fungi, the shoots were contaminated with non-mycelial yeast and/or bacterial infection found in 30.4%, 32%, and 20.8% for upper, middle, and basal sections of the shoots, respectively.

Seasonal changes in the species composition of fungi colonizing the Manchurian apricot shoots were investigated using 10 trees. The trees were relatively young, self-seeded, fruiting plants in satisfactory conditions. The twigs were collected in the beginning of the growing season when the flower buds were yet swollen (30 April 2015), by the end of new shoot growth (8 June 2015), at the end of summer after fruiting (28 August 2015), and during the dormant period (13 November 2015). The highest proportion of shoots colonized by fungi was found in the dormant period (82%) and the lowest one was in the green new shoots (8%) (Table 4). Interestingly, the twigs collected in the beginning of the growing season were less infected with fungi (48%) than the dormant twigs collected after the end of the vegetation period (82%). The highest fungal diversity was also found in November when 11 species were isolated. In addition to the earlier found species, there

were *Au. pullulans*, *C. herbarum*, and *Gliocladium* sp.; however, no strain of *F. oxysporum* was found (Table 5). Other new species isolated in that vegetation period were *T. myrti* (the beginning of the growing season) and *S. strictum* (after fruiting). The dominant and frequent species in all collections of the year was *N. quercina*. Other frequent species were *D. eres* (the beginning of the growing season) and *A. alternata* (after fruiting). Other fungi were rare or occasional species (Table 5).

**Table 4.** Fungal colonization of the Manchurian apricot shoots during the vegetation period of 2015.

| | Phenological Stage (Date of Collection) [1] | | | |
|---|---|---|---|---|
| | **Beginning of Growing Season (30 April)** | **End of New Shoot Growth (8 June)** | **End of Summer (28 August)** | **Dormancy (13 November)** |
| Number of tested trees/shoots | 10/50 | 10/50 | 10/50 | 10/50 |
| Total infected shoots, % | 48 | 8 | 26 | 82 |
| Infected upper sections: | | | | |
| shoots, % | 18 | 0 | 10.4 | 46 |
| buds, % | 2.4 | 0 | 2.1 | 14.6 |
| nodes, % | 15 | 0 | 8.5 | 20.8 |
| internodes, % | 5 | 0 | 0 | 16 |
| Infected middle sections: | | | | |
| shoots, % | 26 | 2 | 12.5 | 40 |
| buds, % | 5 | 0 | 6.4 | 14 |
| nodes, % | 15 | 2 | 8.3 | 18 |
| internodes, % | 2.5 | 2 | 4.3 | 26 |
| Infected basal sections, shoots % | 26 | 8 | 10.4 | 46 |

Notes. Five shoots were collected from each tree (self-seeded, fruiting plants) and surface-sterilized using 70% ethanol for 2 min and 0.2% mercury chloride for 10 min. [1] Brief characteristics of the plant and shoots at the time of collection: Beginning of growing season—flower buds are swollen, wooden twigs are of the last year's growth. End of new shoot growth—young green shoots have completed growth or are close to growth completion. End of summer—completion of fruiting (fruits have dropped off), new shoots are lignifying, buds have formed on this year's growth. Dormancy—leaves have dropped off, transition to dormancy under low temperature, the shoots of this year's growth are completely lignified.

**Table 5.** Species composition and a frequency of occurrence of fungi isolated from the Manchurian apricot shoots during the vegetation period of 2015.

| | Phenological Stage (Date of Collection) | | | |
|---|---|---|---|---|
| **Fungi** | **Beginning of Growing Season (30 April)** | **End of New Shoot Growth (8 June)** | **End of Summer (28 August)** | **Dormancy (13 November)** |
| *Alternaria alternata* | − | − | F | O |
| *A. tenuissima* | − | − | − | O |
| *Aureobasidium pullulans* | − | − | O | R |
| *Cladosporium cladosporioides* | − | − | − | O |
| *C. herbarum* | − | − | − | O |
| *Diaporthe eres* | F | − | − | O |
| *Epicoccum nigrum* | − | − | − | O |
| *Gliocladium* sp. | − | − | − | O |
| *Sarocladium strictum* | − | − | O | − |
| *Nothophoma quercina* | F | + | D | F |
| *Tripospermum myrti* | O | − | − | − |

See Tables 3 and 4 for abbreviations and other details.

The sporadically infected young shoots (8%) collected in June 2015 were found to contain fungi only in the lower (basal and middle) sections (Table 4). The fungal isolates were limited only to *N. quercina* (Table 5). The distribution of fungi among different levels (upper, middle, basal) of shoots was more uniform in other phases of the vegetation period. Nodes were found to be more infected by fungi, though a high proportion of buds and,

especially, internodes were also infected in November (Table 4). It is worth noting that we did not detect any infection in swollen flower buds in the spring (results not shown).

Additional information on the mycobiota of the Manchurian apricot shoots came from the experiments with the apricot explants used for in vitro culture initiation. Table 6 shows the list of fungi isolated from the explants. The explants which consisted of a bud with a fragment of the stem were infected more often than isolated buds without scales. Fragments of young green shoots with nodes were also often contaminated with fungi that can be explained by a larger size of the fragment and the mild method applied for their surface sterilization. *Nothophoma quercina* was the most frequent fungus isolated from the explants derived from shoots at different stages of development (during both the dormant and growing periods). *Cladosporium cladosporioides* and *A. alternata* were usually found in explants during shoot dormancy or in the beginning of shoot growth (bud swelling) that is in agreement with their rare occurrence in the actively growing shoots after surface sterilization (Tables 3 and 5). *Tripospermum myrti* was once isolated during bud swelling. *Diaporthe eres* was found in the explants obtained from growing shoots. Under the mild conditions of surface sterilization (0.2% potassium permanganate, 70% aqueous ethanol or 2-propanol, diluted solutions of the commercial bleach), the degree of fungal contamination significantly increased; the fungi were represented by *N. quercina*, *A. tenuissima*, *D. eres* (results not shown).

**Table 6.** Fungal contamination of the apricot explants used for in vitro culture.

| | Phenological Stage (Date of Collection) | | | | | |
|---|---|---|---|---|---|---|
| | **Dormancy (24 February 2015)** | **Dormancy (25 March 2014)** | **Beginning of Burst of Vegetative Buds (25 April 2014)** | **Swelling of Vegetative Buds (30 April 2015)** | **New Shoot Growth (23 May 2014)** | **New Shoot Growth (30 May 2014)** |
| Total number of explants/number of infected explants | 21/13 | 40/5 | 40/7 | 40/5 | 45/19 | 10/3 |
| Type of explant | Buds with stem segments (bud scales removed) | Buds without scales | Buds without scales | Buds without scales | Segments of green shoots with nodes | Segments of green shoots with nodes |
| Method of surface sterilization [1] | EtOH (2) + HgCl$_2$ (10) | EtOH (2) + HgCl$_2$ (10) | HgCl$_2$ (10) | EtOH (2) + HgCl$_2$ (10) | HgCl$_2$ (5) | HgCl$_2$ (1) |
| Nutrient medium [2] | Modified QL or 1/2QL | Modified MS | QL | QL or Modified QL | QL | QL |
| Isolated fungal species | *N. quercina, A. alternata, C. cladosporioides* | *A. alternata, C. cladosporioides*, sterile mycelium | *N. quercina* | *C. cladosporioides, T. myrti* | *D. eres, N. quercina*, sterile mycelia | *N. quercina*, sterile mycelium |

Notes. [1] Surface sterilization: HgCl$_2$—0.2% mercury chloride with addition of Triton X-100; EtOH—70% ethanol. The number in the brackets indicates time of the treatment in min. [2] Nutrient media: Modified MS—macronutrients, micronutrients, and vitamins according to Murashige and Skoog [57], glucose (2%), sucrose (2%), agar (0.8%), casein hydrolyzate (0.1%), *meso*-inositol (100 mg/L), adenine (40 mg/L), 6-benzylaminopurine (2–16 mg/L), biotin (1 mg/L), folic acid (0.5 mg/L), riboflavin (0.5 mg/L), cobalamin (0.015 mg/L); QL—macronutrients, micronutrients, and vitamins according to Quoirin and Lepoivre [58], sorbitol (2%), agar (0.6%), and 6-benzylaminopurine (3 mg/L) [59]; Modified QL—Quoirin–Lepoivre macronutrients with a modified composition of micronutrients, sucrose (3%), agar (0.6%), glycine (2 mg/L), thiamine (2 mg/L), nicotinic acid (1 mg/L), 6-benzylaminopurine (0.5 mg/L), and indole-3-butyric acid (0.04 mg/L) [59]; Modified 1/2QL—similar to Modified QL with macronutrients diluted to half strength, sucrose (2%), indole-3-butyric acid (0.2 or 0.4 mg/L), no 6-benzylaminopurine added [59].

In the experiment with cut unsterilized shoots extracted with water, the 100-fold dilution resulted in the observation of *A. alternata* colonies in 100%. After the 1000-fold dilution, *N. quercina, C. cladosporioides, T. myrti*, and a sterile mycelium were isolated (results not shown). Dilutions of the water extract of the surface-sterilized shoots revealed no fungal colonies.

## 4. Discussion

### 4.1. Diversity of Fungi Associated with Apricot Species

According to the U.S. National Fungus Collections Fungus–Host Database [60], *P. mandshurica* is a host for 20 species of fungi. Three other apricot species (*Prunus* sect. *Armeniaca* (Scop.) Turcz.), *P. armeniaca* L., *P. mume* (Siebold) Siebold & Zucc. and *P. sibirica* L.,

are reported to be hosts for 365, 125, and 4 fungal taxa, respectively. These big differences in numbers mostly reflect different degrees of exploration of the apricot species as a consequence of their unequal economic value. As a result, the northern apricot species (*P. mandshurica* and *P. sibirica*) remain much less studied for the fungi associated with the plants.

The young and annual shoots of *P. mandshurica* in our study were found to contain thirteen fungal species (excluding sterile mycelia). Twelve of them were identified to a species level (*A. alternata*, *A. tenuissima*, *Au. pullulans*, *C. cladosporioides*, *C. herbarum*, *D. eres*, *E. nigrum*, *F. graminearum*, *F. oxysporum*, *N. quercina*, *S. strictum*, and *T. myrti*) and one to a genus level (*Gliocladium* sp.), all belonging to anamorphic ascomycetes (Supplementary Materials, Table S1). None of these fungi have been reported for *P. mandshurica* and *P. sibirica* with an exception, when investigated apricot trees from the Russian Far East were not attributed to a particular species [35]. For the two more studied apricot species (*P. armeniaca* and *P. mume*), the Fungus–Host Database includes the following fungal taxa which are also reported in our study: *A. alternata*, *A. tenuissima*, *Au. pullulans*, *D. eres*, *F. oxysporum*, species of *Cladosporium*, *Epicoccum*, *Fusarium*, *Gliocladium*, and *Phoma* [60]. Thus, our study supplements the list of fungi not only for *P. mandshurica*, but also for all species of apricots with particular interest being the findings of *N. quercina* and *T. myrti*. It should be noted this is the first report of *D. eres* in the Amur region and *N. quercina* in Russia.

*4.2. Ecological Features, Distribution, and Potential Pathogenicity of the Fungi Associated with the Manchurian Apricot*

The species found in the Manchurian apricot shoots are considered to be saprotrophic fungi (Supplementary Materials, Table S1). They are widely distributed in the soils of the Russian Far East [35]; some of them have been isolated from soils of Blagoveshchensk city (*A. alternata*, *C. cladosporioides*, *C. herbarum*, and *F. oxysporum*) [61]. Some species have been also found in the roots and rhizosphere of fruit trees including apricot (*A. alternata*, *C. herbarum*, *F. oxysporum*, and *Gliocladium roseum* Bainier) [35]. The revealed fungal species can be partly determined by the medium used in our study for fungal isolation. Czapek's agar is synthetic and contains inorganic nitrogen; thus, all isolated fungi were able to utilize inorganic nitrogen for growth. However, the fungal diversity obtained in this study for the *P. mandshurica* annual shoots shares many fungal species, particularly the frequent ones, with those described in the literature for the genus *Prunus* and fruit trees of the *Rosaceae* family and isolated on other nutrient media (potato-dextrose agar and malt-extract agar). Among the fungi isolated from apparently healthy shoots of the fruit trees in Poland [28], the most frequent was *A. alternata*. The researchers also found the fungal species isolated from the Manchurian apricot shoots: *A. tenuissima*, *Au. pullulans*, *C. cladosporioides*, and *E. nigrum* for the healthy shoots, plus *F. oxysporum* and *C. herbarum* for diseased shoots [28]. Unfortunately, the authors showed the fungal composition for the whole set of the fruit trees tested (apple, pear, cherry, plum, apricot, and peach). However, they reported significant colonization of the *P. armeniaca* shoots by *Diaporthe* spp. Similarly, *Alternaria* was the dominant genus in shoots, leaves, and roots of *Prunus avium* grafts from Hungary, followed by different *Fusarium* species and *Epicoccum* sp. [24]. Asymptomatic one- and two-year-old twigs of *Prunus persica* in Uruguay were mostly infected by *Alternaria* spp. and *Aureobasidium* spp., although *Epicoccum* spp. and *Diaporthe* spp. were isolated only from two-year-old twigs [26]. Meanwhile, *Alternaria* sp. was a rare fungus in *Prunus africana* from Cameroon, which was found in the leaves but not in the stems of the plant; more abundant *Fusarium* spp. were also isolated from leaves and roots only; the stems of the plant were infected by different fungi among which *Cladosporium* sp. and *Gliocladium* sp. are to be mentioned [21]. Despite the prevalence of the soil-associated saprotrophic fungi in the Manchurian apricot shoots, we did not isolate *Aspergillus* P. Micheli ex Haller and *Penicillium* Link., which are species common for the soils of Blagoveshchensk city [61], and have been also found in shoots of the fruit trees [21,26,28,62] and other plants [63].

Shoots of the old apricot trees were more infected than the younger trees (Tables 2 and 4) as it can be expected from the sanitary conditions of the trees. The difference in fungal infection between the older and younger trees was especially pronounced for the young newly grown shoots: all tested shoots were colonized, and the species composition was much more diverse for the older trees in June 2014 (Tables 2 and 3) while the colonization level was low for shoots collected from the younger trees in June 2015 with only *N. quercina* isolated (Tables 4 and 5). This discrepancy may be explained by the differences in the age and conditions of the trees and, also, by different sterilizing agents used in both experiments. The two-step surface sterilization with ethanol and mercury chloride used in the experiment of 2015 is a much more severe treatment than the mild sterilization with sodium hypochlorite used in June 2014. In any case, the experiments imply an internal location of *N. quercina* in the shoots.

*Nothophoma quercina* was a frequently isolated fungus in our study; it was found in the dormant twigs as well as in the growing and lignifying shoots (Tables 3, 5 and 6). The association of the fungus with basal sections of shoots (results not shown) and its frequent occurrence in the beginning of the growing season support its endophytic origin. *Nothophoma quercina* appears to begin shoot colonization early and invades young forming shoots from the parental twigs of the previous year.

With the exception of two species (*T. myrti* and *N. quercina*), all other fungi have been reported as endophytes [64]. In our study, the fungi *A. alternata*, *C. cladosporioides*, and *F. oxysporum* were mostly found on the surface of the twigs (see unsterilized shoots in Table 3, the experiment with the unsterilized shoots extracted with water in Section 3.4). On the other hand, the frequent occurrence of *A. alternata* in the twigs after surface sterilization (Tables 3, 5 and 6) may indicate its probable internal location as well.

Distribution of fungi between parts of shoots revealed their frequent association with nodes and buds. While we did not investigate fungal distribution in the shoot tissues, it has been reported that the predominant localization of endophytic fungi is in the dead outer bark of different tree species [65,66]. Inner parts of the buds are mostly free of fungi, while fungi were frequently isolated from the bud scales (Table 2). Similarly, endophytic fungi have been found to be localized in the scale tissues of Scots pine buds [67].

Most of the species (*A. alternata*, *A. tenuissima*, *Au. pullulans*, *C. cladosporioides*, *C. herbarum*, *E. nigrum*, *F. oxysporum*, and *S. strictum*) can be also facultative/opportunistic plant pathogens (Supplementary Materials, Table S1). The *Alternaria* species are associated with many plant diseases [39,68]. As reported for *Prunus laurocerasus* L. in Slovakia [13], *A. alternata* was a causal agent of leaf spots on the tree, and *F. oxysporum* was also often isolated from the plant's diseased leaves and twigs. *Fusarium oxysporum* is a well-known vascular wilt pathogen with a number of races designated [69], while its endophytic isolates have been also reported [70]. A pathogen of cereals, *F. graminearum*, was found only in the unsterilized young shoots of the apricot (Table 3). While the finding was occasional, there is evidence of isolation of the *F. graminearum* species complex from plants of different families besides cereals [71,72]. *Nothophoma quercina* has been found to be associated with cherry leaf spot disease of *P. avium* in China [73]. The fungus was detected from stem cankers on *Fraxinus chinensis* and leaf necrosis on *Ilex cornuta* [74]. Leaf spot diseases can be also caused by *E. nigrum* [75,76] and *S. strictum* [77,78] in different plants, whereas a large number of *Epicoccum* strains were isolated from symptomless leaves [79]. The most pathogenic species among the isolated fungi seems to be *D. eres* which has been shown to be responsible for dieback in *Prunus persica* in China [20,80].

*Tripospermum myrti* was also an occasional, however, interesting finding. The fungus is considered as a rare species for the Russian Far East since it has been reported so far only for the Zeysky State Nature Reserve in the Amur region [81]. In the case of the Manchurian apricot, *T. myrti* was isolated from the shoots in the beginning of the growing season (Tables 5 and 6), and also on the dormant unsterilized shoots extracted with water. The fungus is considered as a foliar epiphyte of living trees; however, its conidia are often found in streams suggesting alternation between its terrestrial and aquatic habitats [82]. *Tr: T. myrti*

*mum myrti* has been found on twigs of several deciduous and coniferous trees [83] and isolated from the needles of *Picea mariana* after surface sterilization [84], which evidences the possible endophytic location of the fungus. Our results for the Manchurian apricot, i.e., isolation of *T. myrti* from visually healthy shoots after surface sterilization, supports the endophytic development of the fungus. While the pathogenicity of *T. myrti* for plants is unknown, it was found as a dominant species in apple sooty blotch complex [85].

A positive effect of the plant colonizing fungi should be taken into account as well: some species isolated in this study (*E. nigrum*, *Au. pullulans*, *C. cladosporioides*, *S. strictum*) have been reported to be useful in the biocontrol of different fungal pathogens and diseases [86–92]. Investigations of the strains inhabiting apricot shoots in terms of fungal antagonism and pathogenicity will provide insights in the contribution of the fungi in the wellbeing of this host plant.

### 4.3. Phylogenetic Analysis of the Pycnidial Fungi

Two pycnidial fungi (*D. eres* and *N. quercina*) were the most frequent species isolated from the Manchurian apricot shoots (Tables 3, 5 and 6). According to the literature data, the species *D. eres* was found in association with many plants including the genus *Prunus* (*P. avium*, *P. cerasus*, *P. cornuta*, *P. davidiana*, *P. domestica*, *P. lannesiana* f. *sekiyama*, *P. mume*, *P. persica*, *P. sargentii*) in USA, Australia, New Zealand, Bulgaria, Greece, Italy, Pakistan, and the countries of east Asia: China, Korea, and Japan [46,60,80]. This fungus is known as a very polymorphic species; its different isolates can form separate clades on the phylogenetic trees depending on a locus used for phylogenetic analysis [46,80]. These clades were treated early as distinct species [46]. Since the designation of the *Diaporthe eres* species complex [46], at least thirty species have been assumed to be its members. However, the investigations on intraspecific relationships in the *D. eres* species complex have revealed the recognition of multiple species in this complex to be redundant [93–99]. Thus, all those thirty *Diaporthe* species, including *D. castaneae-mollisimae* (S.X. Jiang & H.B. Ma) Udayanga, Crous & K.D. Hyde (DNP128), and *D. longicicola* Y.H. Gao & L. Cai (CGMCC 3.17089), which are closely related to the isolates MF-Pm2a, MF-Pm3a, and MF-Pm4a in our study (Figure 1), as well *D. biguttusis* Y.H. Gao & L. Cai (CGMCC 3.17081), *D. mahotocarpus* (Y.H. Gao, W. Sun & L. Cai) Y.H. Gao & L. Cai, (CGMCC 3.15181), and *D. ellipicola* Y.H. Gao & L. Cai (CGMCC 3.17084), which are, respectively, closely related to the isolates MF-Pm1a and MF-Pm5a in our study (Figure 1), are to be considered as synonyms of *D. eres* [97]. Despite that the strains in our study have differences in nucleotide sequences and form distinct subclades in the phylogenetic tree, these subclades are united in the highly supported clade (MLBS 100%, MPBS 100%) which corresponds to the species *D. eres* with similar morphological features.

*Nothophoma quercina* is also a polymorphic species: in all phylogenetic trees (Figure 2), the representative and studied *N. quercina* strains were intermixed with the type and representative strains of *N. brennandiae*, *N. pruni*, and *N. spiraeae*. Thus, we consider four species (*N. brennandiae*, *N. pruni*, *N. quercina*, *N. spiraeae*) and their strains as members of the *N. quercina* species complex. However, *N. spiraeae* could not be distinguished morphologically [100] and phylogenetically from the *N. quercina* strains. It leads us to the conclusion that the recognition of *N. spiraeae* in this complex is redundant. Meanwhile, strains of *N. brennandiae* and *N. pruni* can be distinguished from *N. quercina* morphologically: pycnidia of *N. brennandiae* have setose conidiomata with up to four ostioles [101], while in *N. quercina* conidiomata are glabrous with a single ostiole [100]; conidia of *N. pruni* are larger and hyaline [73], whereas conidia of *N. quercina* turn brown with age [102]. Despite the *N. quercina* isolates in our study differing from each other in molecular phylogenetic features, they possess the same morphological features (Supplementary Materials, Morphological Characterization of Fungi, Figure S7) which correspond to the ex-type *N. quercina* strain (CBS 633.92) [103].

## 5. Conclusions

Young and annual shoots of *P. mandshurica* are inhabited by micromycetes of different taxonomic groups. All isolated species belong to ascomycetes; however, they are diverse in family representation (thirteen species of nine families). Three genera (*Alternaria*, *Cladosporium*, and *Fusarium*) are represented by two species. The isolated fungi are all widespread or even ubiquitous in distribution, and saprotrophic in nutrition. Most of the colonizing fungi are known to be potential endophytes, which is supported by their internal localization in the *P. mandshurica* annual shoots as well. *Nothophomaquercina* and *T. myrti* are found to be potential endophytes of an apricot speciesfor the first time. Identification of *N. quercina* isolates by both multilocus phylogenetic analysis and morphological features providesthe study as a first report of this fungus in Russia.Molecular analysisrevealedredundancy of some species within the genus *Nothophoma*. According to the literature data, the isolated fungal species can also be pathogenic. Since we used visually healthy shoots of the Manchurian apricot, the pathogenicity of the isolated strains towards the plant host is yet to be tested.

**Supplementary Materials:** The following supporting information can be downloaded at: https: //www.mdpi.com/article/10.3390/d14121124/s1, Morphological Characterization of Fungi Isolated from Shoots of *Prunus mandshurica* (Maxim.) Koehne in Blagoveshchensk (Amur Region, Russia) with Figures S1–S8. Figure S1. *Alternaria alternata*: colonies on Cz (**a**), PSA (**b**), and OA (**c**) after 10 d of growth; conidiophores and chains of conidia (**d**); conidia (**e**); Figure S2. *Alternaria tenuissima*: colonies on Cz (**a**), PSA (**b**), and OA (**c**) after 10 d of growth; conidiophores and chains of conidia (**d**); conidia (**e**); Figure S3. *Diaporthe eres* (MF-Pm-1a): colonies on PSA (**a**), OA (**b**), and Cz (**c**) after 14 d of growth; pycnidia on PSA (**d**); alpha and beta conidia on PSA (**e**); Figure S4. *Epicoccum nigrum*: a colony on Cz after 14 d of growth. The reverse side is shown on (**b**); Figure S5. *Fusarium graminearum*: colonies on Cz (**a**,**b**), PSA (**c**,**d**), and OA (**e**,**f**) after 14 d of growth; Figure S6. *Fusarium oxysporum*: colonies on Cz (**a**,**b**), PSA (**c**,**d**), and OA (**e**,**f**) after 14 d of growth; Figure S7. *Nothophoma quercina* (MF-Pm-13a): colonies on PSA (**a**), OA (**b**), and Cz (**c**) after 14 d of growth; pycnidia on PSA (**d**); conidia on PSA (**e**); Figure S8. *Tripospermum myrti*: colonies on Cz (**a**), and conidia in different planes (**b**–**e**); Table S1. A list of fungal species isolated from the Manchurian apricot shoots (*P. mandshurica*) in this study and their known ecological features.

**Author Contributions:** Conceptualization, E.V.N.; methodology, E.V.N., L.P.S. and M.M.G.; validation, A.V.A. and M.M.G.; formal analysis, M.M.G. and L.P.S.; investigation, E.V.N., L.P.S., M.M.G., A.V.A., L.Y.K. and L.M.P.; resources, L.M.P.; data curation, M.M.G.; writing—original draft preparation, E.V.N., L.P.S. and M.M.G.; writing—review and editing, M.M.G. and A.V.A.; visualization, E.V.N. and M.M.G.; supervision, E.V.N. and L.M.P.; project administration, E.V.N.; funding acquisition, E.V.N., M.M.G. and A.V.A. All authors have read and agreed to the published version of the manuscript.

**Funding:** This research was funded by the Ministry of Science and Higher Education of the Russian Federation (projects No. AAAA-A20-120042090002-0, 122040800086-1, and 121032300081-7) and by the Russian Science Foundation (project No. 19-76-30005).

**Institutional Review Board Statement:** Not applicable.

**Data Availability Statement:** The new generated sequences are deposited in GenBank (https://www. ncbi.nlm.nih.gov/nuccore/) under accession numbers listed in Table 1. All other data is contained within the article or Supplementary Materials.

**Conflicts of Interest:** The authors declare no conflict of interest.

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
