# Peer review of "Diversity of Endophytic Fungi in Annual Shoots of Prunus mandshurica (Rosaceae) in the South of Amur Region, Russia"

_diversity, doi:10.3390/d14121124_

Round 1

Reviewer 1 Report

Dear authors,

Your manuscript entitled: "Fungi of annual shoots of Prunus mandshurica (Rosaceae) in the south of Amur Region, Russia has been reviewed. I noticed a number of grammatical errors throughout the text but there are many more. Therefore, I invite you to send the new manuscript to a native speaker to prepare a better version. Regarding to this manuscript the following points and suggestions in to the text (attached pdf file) should be considered by the respected authors.

1-The title: the title of the manuscript is not proper. It is better to choose a more appropriate title based on the study (biodiversity and endophytic fungi).

2-Abstract: this part of article should state briefly the introduction (host important), materials and methods, results and discussion (new data of this study for Russia or the world), So this section should be rewritten.

3-Introduction: Most of the introduction section is about endophytic and epiphytic fungi. But it is better to emphasize about the variety of endophytic fungal species that have been reported from Prunus spp..

4-Materials and Methods: This part should be divided two sections (with two headings): fungal isolations from young green shoots and fungal isolations from dormant twigs. This section needs additional explanation especially in the field of the fungal isolations, purifications and morphological identification of the isolates.

 5- Results: Presenting the results seems a bit confusing. Therefore, the order of the headings written in the results section should be consistent with the Materials and Methods (The order of headings and their descriptions in the results section should be based on the order specified for Materials and Methods). On the other hand, the framework of the manuscript should be uniform and the results and discussion should be written according to the headings of materials and methods. It is mentioned that other isolates obtained during this study were identified using ITS. Although this does not work precisely for all isolates from different taxa, the obtained molecular results are not explained in this part of the results. By the way, no explanation has been given about the morphological features and references to identify the isolates. In the phylogenetic tree, the clade of D. eres seems very complex and the isolates obtained in this study are placed in different groups. So, it will be difficult to interpret the results.

6- Discussion: More articles (as references, references that are closely related to this study) should be reviewed for this part of the manuscript. It is better that the new reports of any species from Russia or the world are clearly stated on the studied tree.

Therefore, this manuscript requires Major Revision and then it should be reviewed again in revised version.

Good luck.

Reviewer 2 Report

Manuscript: diversity-2009530

Fungi of annual shoots of Prunus mandshurica (Rosaceae) in the south of Amur Region, Russia.

In general it is nice work and well written manuscript, except for a few grammatical mistakes, e.g. “did not found” should be “did not find” in several places in the manuscript. The authors should also use “fungal infection” instead of “fungal contamination”, perhaps with the exception when the plant material have not been surface disinfected. Materials & Methods should be more systematic and both Results and Discussion needs shortening and precision.

Specific comments:

M&M:

It there an additional table S1? (L164)

It is a bit confusing with the different experiments using different sterilization methods. It needs to be more systematic explaining the different experiments separate and in chronological order.

The paragraph (L116-119) does not make sense. The authors have to describe the experimant. What is explant?

Why was Czapek’s agar used for all fungal genera?

Alternaria seem to be the most abundant genus (141/331). Why did you not use a more dedicated reference for identification that Compendium of soil fungi? It has two Alternaria species and all your isolates fitted those descriptions? You could have used Woudenberg et al. (2013), which is free. https://www.ncbi.nlm.nih.gov/pmc/articles/PMC3713888/pdf/simycol_75_1_004.pdf. Were any of the many Alternaria isolates sequenced? And why not, if “no”?

Now that the authors have most of their isolates on Czapek agar, it would be nice to see the cultures on this medium as well as PSA and OA – is that possible?

It would probably help the reader (those that are not plant pathologists) if the authors could make a drawing/figure showing from where the shoots (distal, middle, proximal) originated and how they relate to each other.

Results:

Combine Tables 1 and 2.

Add reference that state that N. quercina is a species complex.

Table 8 should be in supplementary material

Discussion:

The authors can only compare two or more isolates if they have both/all in culture under the same conditions (L533-542). It may be that they are all the same species, but to prove it needs all existing isolates in culture at the same time.

Discussion needs to be shorter and more concise.

Layout:

Add an extra line between table notes and the main text, e.g. L340, L 359, L402, L421

Round 2

Reviewer 1 Report

Dear Authors,

I thank you for making the necessary changes and corrections in the text of the manuscript. It seems that by making the requested suggestions and changes, the revised format of the manuscript has now been improved and is ready for acceptance.

Good luck and best wishes to you.